# Gender Gap in Healthcare Worker–Patient Communication during the Covid-19 Pandemic: An Italian Observational Study

**Vitale Elsa**

Department of Mental Health, Local Healthcare Company Bari, Via X Marzo, 43, 70026 Modugno, BA, Italy; vitaleelsa@libero.it

**Abstract:** The value of the healthcare worker–patient communication has been well demonstrated and validated in several studies evidencing its relation to positive patient health outcomes, including better care response, simpler decision-making, better patient psychological well-being, and, therefore, considerable patient care satisfaction. The present study purposed to assess how patients perceived healthcare worker–patient communication during the COVID-19 pandemic and whether there were any gender-related differences among participants. From March 2020 to April 2020, an online questionnaire was administered to those who declared a patient's condition in this period. The data considered included data on gender and a Quality of Communication questionnaire (QOC). A total of 120 patients were recruited online. Of these, 52 (43.33%) were females and 68 (56.67%) were males. Significant differences were recorded between females and males in the QOC questionnaire as regards Item no. 2 ($p = .033$), Item no.6 ($p = 0.007$), Item no.11 ($p = 0.013$), Item no.12 ($p = 0.003$), Item no.13 ($p = 0.002$), Item no.15 ($p = 0.008$), and Item no.16 ($p = 0.037$), respectively. The potentially different elements between the two sexes considered were assessed in: Component 1: the need to be completely informed about their own health condition, and Component 2: the need to receive authentic and sincere communication from the healthcare worker involved. In light of the present findings, it has emerged that male patients seemed to be more active and positive in effective healthcare worker–patient communication.

**Keywords:** communication; gender gap; healthcare worker; patient

## 1. Introduction

The value of the healthcare worker–patient communication has been well demonstrated [1–4] and validated in several studies evidencing its relation to positive patient health outcomes [5–7], including better care response, simpler decision-making [8], better patient psychological well-being, and, therefore, considerable patient care satisfaction [9,10]. Several studies have suggested that the quality of healthcare worker–patient communication is currently low [11,12] and that healthcare teams are often unconscious of the desires of their patients [11,12]. In a study conducted in Germany [13] including patients suffering from multiple sclerosis, as well as in an Australian study [14] enrolling patients with ductal carcinoma in situ, it was highlighted that many patients were dissatisfied with their communications with the entire healthcare team; they required more information about their disease in all its aspects than they were given. Moreover, a good and equilibrated communication approach presumes an unprejudiced input from healthcare worker and patient [15,16]. This approach varies from a paternalistic communication strategy in which the healthcare worker influenced the entire communication process by making decisions [15,17]. The informed and assigned decision-making between healthcare worker and patients represents the significant topic in the ideal style [18]. However, this approach has been mostly recommended in Western countries. In Southeast Asian countries, an unusual communication approach may be more suitable,

because social grading is more important and more generally agreed upon. An essential feature of this ranking system is the regard due to someone recognized as being of a higher social condition. Two other aspects with a significance for healthcare worker–patient communication and in which Southeast Asian societies vary from most Western cultures are, first, a reduced autonomy of the members of a social group, which Hofstede has identified as collectivism (as opposed to individualism), and second, the adoption of traditional medicine, which is also impacted by collectivism [19–21]. These social attributes may develop a one-way or paternalistic communication approach in which the healthcare worker controls the dialogue. It should be also considered that, in Asian societies, patient–healthcare worker communication has generally involved a relationship in which the healthcare worker demonstrates a caring condition rather than communication between two sides that are on the same level, as in the Western societies [21,22]. Additionally, the literature suggests that healthcare worker–patient communication also depends on patients' educational levels [23–26]. Moreover, evidence has highlighted associations between health care workers' use of effective communication approaches and the extent to which patients are satisfied with healthcare outcomes, thanks to the communication of healthcare workers' expertise in such a way as to allow patients' participation in their healthcare [27,28]. However, none of the standardized communication protocols or strategies considered the possible existence of a gender gap which could influence communication between patients and healthcare workers.

*Aim*

The purpose of the present study is to assess how patients perceived healthcare worker–patient communication during the COVID-19 pandemic and whether there were any gender-related differences among participants.

## 2. Materials and Methods

### 2.1. Study Design

An observational, national, cross-sectional study using the snowball sampling method was conducted from March 2020 to April 2020.

### 2.2. Data Collection

An online questionnaire was administered to all Italian citizens who were in the "patient" condition in the period considered, in order to assess their perceptions of the quality of their communication with their healthcare workers during the Covid-19 pandemic. No further inclusion or exclusion criteria were considered. The questionnaire was developed through the Google function, as: GOOGLE MODULES. For patient recruitment, only general Facebook and Instagram pages were used. The questionnaire was filled in anonymously and the only important aspect considered for the enrollment was the willingness of the participants to answer the anonymous questionnaire.

### 2.3. The Questionnaire

An online questionnaire was created ad hoc containing data referring to gender (female or male) and then, a Quality of Communication questionnaire (QOC) [27] (Appendix A). The QOC contained a total of 17 items associated with a Likert scale, which varied from 1 to 5, with "1" indicating "never" and "5" indicating "very often".

For each item considered in the QOC, it was possible to assess the perceived quality of healthcare worker–patient communication during the COVID-19 pandemic [29].

### 2.4. Validity and Reliability

The Quality of Communication questionnaire (QOC) was developed by Curtis et al. [30] by assessing the quality of patient–physician communication in palliative care settings. Then, in 1999, the QOC was validated in a cohort of AIDS patients and their physicians [31], and in 2002 [32] it was adopted in a qualitative study including focus groups of patients suffering from AIDS, cancer, or COPD. The QOC questionnaire has been administered in several parts of the world for different care settings and also to evaluate several patients' condition groups and their perceptions of communication quality from their related healthcare workers. For example, in the Netherlands, the QOC was used in end-stage renal disease patients on dialysis [33] as well as in patients with advanced COPD, chronic heart failure, or chronic kidney disease [34]. In the USA, the QOC was administered to chronic obstructive pulmonary disease (COPD) patients receiving palliative care [31,32] and in Germany to patients suffering from multiple sclerosis [13]. Additionally, in the USA [35–37] and in Canada [38] the QOC was used to investigate the quality of communication between physicians and severely ill patients with a ≥50% rate of mortality. To date, there is no validated version of the QOC in Italian. However, in the current literature there was only a personal translation of the QOC questionnaire in Italian, not yet validated [29]. The QOC instrument was considered a monofactorial analysis in the assessment of perception of communication quality between patient and healthcare worker. All the QOC items reported good internal consistency ($\alpha$ = 0.91 and $\alpha$ = 0.79, respectively), and significant associations ($p < 0.01$) were demonstrated in the content validity [31,32]. Additionally, in the present study, the internal consistency as regards all the 17 items included was assessed as: $\alpha$ = 0.958.

### 2.5. Data Assessment

Data were all collected in an Excel data sheet and processed with the SPSS, version 20. All the answers received were assessed as distribution curves performing the Shapiro–Wilk and Kolmogorov–Smirnov tests. As the distributions of the variables analyzed did not conform to a Gaussian distribution ($p < 0.001$), the data were presented as frequencies and percentages. *Chi square tests* were also performed to assess any differences in patients' quality perceptions during the pandemic according to the gender variable. All $p$ values < 0.05 were considered statistically significant. Then, all the items showing significant differences between females' and males' perceptions in their communication with their healthcare workers were further explored through exploratory factor analysis using the extraction method from maximum likelihood analysis and the rotation method from varimax with Kaiser Normalization in order to explore any latent factors which could contribute to any differences between females and males in the quality of communication between patient and healthcare worker. The significance level was assessed at 0.05.

### 2.6. Ethical Considerations

The questionnaire was administered only in an online form. Participation was voluntary and no form of personal restitution of the results obtained was involved. All the information collected had no diagnostic purpose and the results were treated anonymously. The present study was approved by the Ethical Committee of Polyclinic in Bari, Italy, with no. 6463/2020.

## 3. Results

A total of 120 patients were recruited online who, during the period considered, experienced the "patient" condition. Of these, 52 (43.33%) were females and 68 (56.67%) were males. As shown in Table 1, significant differences were recorded between females and males in the QOC as regards Item 2 ($p = 0.033$), Item 6 ($p = 0.007$), Item 11 ($p = 0.013$),

Item 12 (*p* = 0.003), Item 13 (*p* = 0.002), Item 15 (*p* = 0.008), and Item 16 (*p* = 0.037), respectively.

**Table 1.** QOC assessment in patients during the COVID-19 pandemic.

| QOC Items/QOC Answers | Never n (%) | Sometimes n (%) | Frequently n (%) | Often n (%) | Very Often n (%) | *p*-Value |
|---|---|---|---|---|---|---|
| **Item no.1** | | | | | | |
| Female | 8 (6.67) | 21 (17.50) | 14 (11.67) | 6 (5.00) | 3 (2.50) | 0.105 |
| Male | 4 (3.33) | 18 (15.00) | 24 (20.00) | 15 (12.50) | 7 (5.83) | |
| **Item no.2** | | | | | | |
| Female | 5 (4.17) | 22 (18.33) | 16 (13.33) | 7 (5.83) | 2 (1.67) | 0.033 * |
| Male | 4 (3.33) | 18 (15.00) | 17 (14.67) | 15 (12.50) | 14 (11.67) | |
| **Item no.3** | | | | | | |
| Female | 8 (6.67) | 15 (12.50) | 15 (12.50) | 10 (8.33) | 4 (3.33) | 0.235 |
| Male | 13 (10.83) | 16 (13.33) | 10 (8.33) | 19 (15.83) | 10 (8.33) | |
| **Item no.4** | | | | | | |
| Female | 7 (5.83) | 10 (8.33) | 19 (15.83) | 8 (6.67) | 8 (6.67) | 0.127 |
| Male | 3 (2.50) | 22 (18.33) | 16 (13.33) | 13 (10.83) | 14 (11.67) | |
| **Item no.5** | | | | | | |
| Female | 5 (4.17) | 9 (7.50) | 22 (18.33) | 12 (10.00) | 4 (3.33) | 0.161 |
| Male | 3 (2.50) | 17 (14.67) | 21 (17.50) | 13 (10.83) | 14 (11.67) | |
| **Item no.6** | | | | | | |
| Female | 8 (6.67) | 5 (4.17) | 20 (16.67) | 14 (11.67) | 5 (4.17) | 0.007 * |
| Male | 3 (2.50) | 20 (16.67) | 15 (12.50) | 17 (14.67) | 13 (10.83) | |
| **Item no.7** | | | | | | |
| Female | 7 (5.88) | 7 (5.83) | 19 (15.83) | 13 (10.83) | 6 (5.00) | 0.093 |
| Male | 4 (3.33) | 19 (15.83) | 16 (13.33) | 15 (12.50) | 14 (11.67) | |
| **Item no.8** | | | | | | |
| Female | 9 (7.50) | 7 (5.83) | 19 (15.83) | 12 (10.00) | 5 (4.17) | 0.572 |
| Male | 9 (7.50) | 17 (14.67) | 21 (17.50) | 13 (10.83) | 8 (6.67) | |
| **Item no.9** | | | | | | |
| Female | 7 (5.83) | 8 (6.67) | 16 (13.33) | 11 (9.17) | 10 (8.33) | 0.313 |
| Male | 12 (10.00) | 18 (15.00) | 17 (14.17) | 15 (12.50) | 6 (5.00) | |
| **Item no.10** | | | | | | |
| Female | 7 (5.83) | 14 (11.67) | 17 (14.17) | 10 (8.33) | 4 (3.33) | 0.065 |
| Male | 24 (20.00) | 19 (15.83) | 13 (10.83) | 9 (7.50) | 3 (2.50) | |
| **Item no.11** | | | | | | |
| Female | 8 (6.67) | 13 (10.83) | 19 (15.83) | 9 (7.50) | 3 (2.50) | 0.013 * |
| Male | 29 (24.17) | 16 (13.33) | 11 (9.17) | 10 (8.33) | 2 (1.67) | |
| **Item no.12** | | | | | | |
| Female | 6 (5.00) | 16 (13.33) | 18 (15.00) | 11 (9.17) | 1 (0.83) | 0.003 * |
| Male | 28 (23.33) | 16 (13.33) | 13 (10.83) | 7 (5.83) | 4 (3.33) | |
| **Item no.13** | | | | | | |
| Female | 9 (7.50) | 7 (5.83) | 25 (20.83) | 9 (7.50) | 2 (1.67) | 0.002 * |
| Male | 9 (7.50) | 26 (21.67) | 13 (10.83) | 12 (10.00) | 8 (6.67) | |
| **Item no.14** | | | | | | |
| Female | 8 (6.667) | 13 (10.83) | 18 (15.00) | 10 (8.33) | 3 (2.50) | 0.400 |
| Male | 15 (12.50) | 16 (13.33) | 15 (12.50) | 13 (10.83) | 9 (7.50) | |
| **Item no.15** | | | | | | 0.008 * |

| | | | | | | |
|---|---|---|---|---|---|---|
| Female | 6 (5.00) | 10 (8.33) | 23 (19.16) | 10 (8.33) | 3 (2.50) | |
| Male | 12 (10.00) | 16 (13.33) | 11 (9.17) | 15 (12.50) | 14 (11.67) | |
| **Item no.16** | | | | | | |
| Female | 8 (6.67) | 13 (10.83) | 16 (13.33) | 14 (11.67) | 1 (0.83) | 0.037 * |
| Male | 23 (19.16) | 19 (15.83) | 9 (7.50) | 13 (10.83) | 4 (3.33) | |
| **Item no.17** | | | | | | |
| Female | 9 (7.50) | 13 (10.83) | 14 (11.67) | 13 (10.83) | 3 (2.50) | 0.124 |
| Male | 15 (12.50) | 16 (13.33) | 12 (10.00) | 11 (9.17) | 14 (11.67) | |

* $p < 0.05$ is statistically significant.

By considering only the items with statistically significant differences between males and females ($p < 0.005$) and further performing the varimax rotation analysis, the potentially different elements between the two sexes considered were assessed, as (Table 2):

1. Component 1: the need to be totally informed about their own health condition;
2. Component 2: the need to receive authentic and sincere communication from the healthcare worker involved.

**Table 2.** Rotated Component Matrix for Exploratory Factor Analysis.

| QOC Items | Component 1 | Component 2 |
|---|---|---|
| Item no.2 | 0.661 | 0.679 |
| Item no. 6 | 0.794 | 0.447 |
| Item no.11 | 0.821 | −0.442 |
| Item no.12 | 0.826 | −0.500 |
| Item no.13 | 0.881 | −0.174 |
| Item no.15 | 0.863 | 0.269 |
| Item no.16 | 0.598 | −0.180 |

Notes: Extraction method was maximum likelihood analysis. Rotation method was varimax with Kaiser Normalization.

## 4. Discussion

The present study aimed to assess how patients perceived healthcare worker–patient communication during the COVID-19 pandemic and whether there were any gender-related differences among participants.

By considering differences in data collected between the patients recruited according to their gender, and then, the hidden components which characterized these differences between them, two significant components were identified as significantly different between males and females. Males seemed to have a major need to be totally informed about their own health condition through authentic and sincere communication with the healthcare worker involved.

It was already known that women and men were radically dissimilar from each other, as the literature describes "gender identity" as the emotion an individual feels by being male or female [39–41]. It is associated with the level to which a person believes that their gender is part of their personality, peculiarities and functions attributed by society to men and women [42]. There are different hypotheses about the recognition of gender identity. The multifactorial theory suggested by Spence [43] describes it according to four essential elements, namely: masculinity and femininity traits according to the categories of the instrumental (masculine) versus the expressive (feminine); gender stereotypes, such as activities, traits, or attributes which are different between men and women; gender functions, such as dominant activities associated with a social role; and attitudes towards gender roles, referring to the evaluation of different functions between men and women [40].

The literature suggests that males and females are characterized by different qualities, capabilities, and tendencies penetrating all age groups, all time periods, and all cultures [44,45]. Such convictions, better identified as patterns, have also been discovered to be more hostile to change, as communality and agency. In this regard, women have been recognized as having a persistent predisposition to be communal in caring for and paying attention to the well-being of others. The conventional woman is described as kind, caring, delicate, energetic, and spiritual. On the other hand, men are described as primarily powerful and instrumental, and as independent, private, determined, vigorous, and strong [46]. Sexual associations were influenced by traditional ideas about sex and gender that linked maleness and masculinity with confidence, hostility, sexual adventurism, and sentimental limitation, and femaleness and femininity with obedience, resignation, sexual humility, and emotional confidentiality. These schemes contributed instructions concerning men's and women's self-presentation and attitudes [47], notified what characteristics were recognized as interesting in sexual or private collaborations [48], and advised the lines conducing individuals through sexual relationships [49,50]. Sex and gender patterns could have negative repercussions on the sexual, relational, and psychological well-being of men and women. For example, Briton and Hall [51] suggested that people reported that women were more nonverbally suggestive and sensitive than men. Moreover, some nonverbal attitudes were considered for females more than for males, such as crying, which is considered as denoting femininity [52,53]. As concerns biological factors, the literature highlights differences in brain structure and functioning by classifying brains into "male" or "female" [54]. In addition, there are few differences in verbal cognitive capabilities between men and women [55,56].

While the differences in verbal cognitive abilities between men and women are few [55,56], several studies have shown sex differences in reading abilities, with females seeming to be more motivated in reading ability [57–60], by increasing with age, too [61,62]. According to this assumption, the performance of individuals could be suggested by expectations to do a given activity well, and on the importance that they give to that activity [63–66]. Since females tend to have a better self-concept in language than males and to value reading tasks more, expectancy-value theory explains their higher performance in that area [61,67–70]. This theory suggests that individuals' inspirational opinions are essential in describing males' lower accomplishments in language attitudes [71]. In this regard, the data presented in this study were in agreement with the current literature, as males who recorded lower language attitudes also needed to receive more authentic and direct communication with their healthcare workers. Therefore, the current study cannot not draw a clear boundary line between males and females in communication management, but it does provide an interesting suggestion in the communication targets between patient and healthcare worker, underlining the need for male patients to receive more direct news. This need may be connected to their gender-related characteristics, which were highlighted by the literature mentioned above. However, the present study has some limitations. First of all, the number of interviewers was not surely representative of the current general population in the patient role. Secondly, data were collected only online and there was no form of iteration with the participants. Furthermore, the major flaw of this study was the sampling and retroactive evaluation of communication, which might have been influenced by patients' memories in their hospital stay. Moreover, the present findings were collected during the COVID-19 pandemic; healthcare worker–patient communication thus may have been influenced by the pandemic period [72,73]. Finally, the data collected were very minimal, as the present study only considered gender and the items of the QOC, and not other socio-demographic characteristics.

## 5. Conclusions

In light of the present findings, it has emerged that male patients needed to receive more direct and authentic communication from their healthcare worker, above all during

the COVID-19 pandemic. Therefore, the present study did not indicate how healthcare workers could communicate with their patients, but only emphasized that healthcare workers should communicate with their patients in a more authentic and comphrensive way, by also considering that male patients had this particular need more than female patients did.

**Funding:** This research received no external funding.

**Institutional Review Board Statement:** The study was conducted in accordance with the Declaration of Helsinki and approved by the Ethical Committee of Polyclinic in Bari, Italy, with no. 6463/2020.

**Informed Consent Statement:** Informed consent was obtained from all subjects involved in the study.

**Data Availability Statement:** The data presented in this study are available on request from the corresponding author.

**Conflicts of Interest:** The author declares no conflict of interest.

## Appendix A. The Quality of Communication Questionnaire (QOC) Administered

| QOC Items |
| --- |
| Item 1: Uses words you understand |
| Item 2: Looks you in the eye |
| Item 3: Includes loved ones in the discussion of therapeutic treatment |
| Item 4: Answers all questions concerning the disease |
| Item 5: Listens to what you have to say |
| Item 6: Takes care of you as a person |
| Item 7: Pays full attention |
| Item 8: Talks about the feelings felt about the worsening of the disease |
| Item 9: Discusses details if you got sick |
| Item 10: Talks about how much time you have left to live |
| Item 11: Talks about what it should be like to die |
| Item 12: Talks to loved ones about how dying could be |
| Item 13: Gets involved in therapeutic discussions about your care |
| Item 14: Asks questions about the important things in life |
| Item 15: Respects the important things in life |
| Item 16: Asks about spiritual and religious beliefs |
| Item 17: Respects spiritual and religious beliefs |

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
