# Peer review of "Gender Gap in Healthcare Worker—Patient Communication during the COVID-19 Pandemic: An Italian Observational Study"

_psych, doi:10.3390/psych4010010_

Round 1
Reviewer 1 Report
The study purposed to assess how patients perceived healthcare worker-patient communication during the Covid-19 pandemic and if there were any gender-related differences among participants. This study is simple. Some comments for the authors to improve its quality.
- In methodology, a section to describe the participants and the sampling method is missing.
- T test should be used for assessing any differences of gender in patients’quality perceptions.
- The sample size is too small to generate concrete results. More participants should be involved in this study.
- The theoretical and practical implications of this study were weak.
Author Response
Dear Reviewer,
thank you for your helpful suggestions.
As attached file I'll send you the revised form
Best Regards
Elsa Vitale
Reviewer 2 Report
Dear Authors, I read your work entitled “Gender Gap in Healthcare Worker-Patient Communication 2 Quality During the Covid-19 Pandemic: An Italian Observational Study” and here I enclose my comments/recommendations:
Overall dynamics of the article: Overall, the strong point of this work is the focus on the communication of healthcare workers with patients in many western countries and more specifically on gender differences.
Title: It would be more tempting to modify the title for greater flexibility of been found by other researchers.
Introduction: The last paragraph of the introduction is almost the same as the abstract. It would be good to modify it. Also, it would be good to make it sound and clear in the introduction why it is so important to do this study. In addition, I would like the research question to be more distinct, even with an albeit manner.
Materials and Methods: I suggest you mentioning more inclusion criteria. The study seems to have weaknesses due to the way the data were collected. Please, provide more data upon the sample collection. I suggest removing the list of questions from the tool completely and if it is necessary to add it as an appendix. It would be good to have patient clinical and socio-demographic characteristics for a more detailed and complete layout of your work.
Validity and Reliability: Some of the information could be summarized in section 2.2 Questionnaire and its more extensive version at the end of the Introduction section.
Results / conclusion: The limitations of the study should be written as there are many that have not been mentioned and affect the final result (e.g., the number of the sample, the absence of citation of the clinical features of the study and especially since differentiation per disease was reported at admission it would be important to report additional data.
General comments: The text has many double spaces, glued characters (e.g. words) to each other without an intermediate space and many typing errors. There is a need for English editing of this work as well.
Thank you!!
Author Response
Dear Reviewr
thank you for your suggestions.
as attached file I'll send you the revised form
Best regards
EV

Reviewer 3 Report
This study, looking at gender differences in healthcare worker-patient communication during COVID-19 provides some insight into how care solutions could be tailored in the future. Unfortunately, the presentation of the paper is inadequate and quite difficult to follow. Specific comments are included below.
Extensive language editing is required. For example "Mars 2020" should be March, "Linker Scale" should be Likert. The manuscript is difficult to read due to the poor English throughout.
Was ethical approval obtained from an ethics board?
The results in Table 1 would be better presented graphically.
The discussion raises some interesting points but there is little discussion about what the findings actually mean in terms of planning for future healthcare. It would be beneficial to include s paragraph about how to apply your findings. Similarly for the conclusion - what do your findings mean?
Author Response
Dear Reviewer
as attached file I'll send you the revised file
Best Regards
EV

Round 2
Reviewer 1 Report
There is one comment left. The abstract should be updated with a sample size of 120 and the results accordingly.
Author Response
As attached file I'll send you the revised version of the manuscript.
Best Regards
Elsa Vitale

Reviewer 2 Report
Dear Authors,
I read your revised work entitled “Gender gap in healthcare worker-patient communication quality during the Covid-19 pandemic: an Italian observational study” and here I enclose my recommendations to you:
- Overall, the manuscript has been improved however some issues remain. There is still a need to improve English and typing errors (eg., statystically significant-line 174).
- Introduction: has improved been enough but there is still some space for improvement on the comments we made upon the clarity of the aims and the rational of this study
- Materials and Methods: No change has been made regarding clinical and socio-demographic characteristics.
- Results: The number of the participants was doubled. Why the Authors did not included the 60 extra participants from the very beginning. What mediated and changed this number?
- I think that Table 1 is a bit tedious to read. I suggest that only Items that have p <.05 should be mentioned along with the question and not the item number to make it easier for the reader.
- Discussion: Καταλήγει στο συμπέρασμα ότι είναι σε ίδιο μήκος κύματος με άλλες μελέτες, “as males, who recorded lower language attitudes also need to receive a more authentically and directly communication with their healthcare workers, too”, όμως δεν μας το σχολιάζουν επαρκώς στον τομέα discussion πως αυτό μπορεί να σχετίζεται με τον COVID-19 μιας και η κεντρική ιδέα είναι η πανδημία COVID-19. Θα ήθελα βελτίωση στη συζήτηση για το ισχυρό σημείο της μελέτης που είναι ο covid-19.
- The Authors conclude that they are on the same wavelength with other studies eg., "….as males, who recorded lower language attitudes also need to receive a more authentically and directly communication with their healthcare workers, too….", but they do not comment enough in the discussion how this may be related to COVID-19 as the central idea is the COVID-19 pandemic. I would suggest again the Authors to improve the discussion since the strength of the study is the Gender gap in healthcare worker-patient communication quality during the COVID-19 pandemic.
Thank you
Author Response
All the changes are reported in the rebuttal before the manuscript attached

Reviewer 3 Report
Thank you to the authors for their revisions. The manuscript has been improved in terms of content and applicability to current healthcare environments. The results are more clear and linked well to the discussion. The language is still an issue and will need to be reviewed as per my comments below.
The Abstract needs significant language editing. The first and second sentences do not make sense.
The introduction needs further language editing, although it has improved. For example the first sentence should read "Has been well demonstrated" and the second sentence should read "Has been validated". Line 23 should read "is low". Line 36 - what is "greatly ranking"? Line 61 - your aim doesn't make sense. It could read "The present study aims to asses...". When talking about literature or other studies, talk about them in present tense not past. For example "literature suggests that...". The tense seems to incorrect in many instances throughout the introduction and the discussion.
Line 174 - spelling error - statistically.
Author Response
All the changes are reported in the manuscript attached
